# Authentication and Traceability Study on *Barbera d'Asti* and *Nizza* DOCG Wines: The Role of Trace- and Ultra-Trace Elements

**Maurizio Aceto** [1,*] , **Federica Gulino** [1] , **Elisa Calà** [1] , **Elisa Robotti** [1] , **Maurizio Petrozziello** [2] , **Christos Tsolakis** [2] **and Claudio Cassino** [1]

1   Dipartimento di Scienze e Innovazione Tecnologica, Università del Piemonte Orientale, viale T. Michel, 11-15121 Alessandria, Italy; federica.gulino@uniupo.it (F.G.); elisa.cala@uniupo.it (E.C.); elisa.robotti@uniupo.it (E.R.); claudio.cassino@uniupo.it (C.C.)
2   CREA Consiglio per la Ricerca in Agricoltura e l'Analisi dell'Economia Agraria, Centro di Ricerca Viticoltura ed Enologia, via Pietro Micca, 35-14100 Asti, Italy; maurizio.petrozziello@crea.gov.it (M.P.); christos.tsolakis@crea.gov.it (C.T.)
*   Correspondence: maurizio.aceto@uniupo.it; Tel.: +39-0131-360265

**Abstract:** *Barbera d'Asti*—including *Barbera d'Asti superiore*—and *Nizza* are two DOCG (Denominazione di Origine Controllata e Garantita) wines produced in Piemonte (Italy) from the *Barbera* grape variety. Differences among them arise in the production specifications in terms of purity, ageing, and zone of production, in particular with concern to *Nizza*, which follows the most stringent rules, sells at three times the average price, and is considered to have the highest market value. To guarantee producers and consumers, authentication methods must be developed in order to distinguish among the different wines. As the production zones totally overlap, it is important to verify whether the distinction is possible or not according to metals content, or whether chemical markers more linked to winemaking are needed. In this work, Inductively Coupled Plasma (ICP) elemental analysis and multivariate data analysis are used to study the authentication and traceability of samples from the three designations of 2015 vintage. The results show that, as far as elemental distribution in wine is concerned, work in the cellar, rather than geographic provenance, is crucial for the possibility of distinction.

**Keywords:** ICP-MS; trace elements; wine; *Nizza*; *Barbera*; authentication

## 1. Introduction

*Barbera d'Asti* DOCG and *Nizza* DOCG are two high-quality wines produced in Piemonte (Italy) from *Barbera* grape variety (*Vitis vinifera*), an autochthonous vine cultivated in that region since 16th century. The designation *Barbera d'Asti* was firstly labelled as DOC (Denominazione di Origine Controllata) in 1970, approved with DPR 09/01/1970 [1] and later on as DOCG (Denominazione di Origine Controllata e Garantita) in 2008, approved with DM 08.07.2008 [2]; the designation involved 116 communes in the Asti province and 51 communes in the Alessandria province for a total surface of 53 km$^2$ (5300 Ha), of whom nearly 40 km$^2$ (4000 Ha) claimed in 2018. The DOCG designation also provided the possibility of using an additional, finer specification as *Barbera d'Asti superiore* for wines produced with minimum ageing of 14 months, 6 of whom in barrique; moreover, there was the possibility of adopting the three specific labelling *Barbera d'Asti superiore sottozona Colli Astiani*, *Barbera d'Asti superiore sottozona Nizza* and *Barbera d'Asti superiore sottozona Tinella* in the case of wines produced, within the whole *Barbera d'Asti* area, in the three corresponding geographic sub-zones, considered as the more suitable in terms of quality. Recently the *Barbera d'Asti superiore sottozona*

*Nizza* has been elevated to the rank of a new DOCG [3] called simply *Nizza*, according to more severe rules that included production in only 18 communes inside the Asti province, located around Nizza Monferrato (Figure 1), for a total area under vines of 7.2 km$^2$ (720 Ha), of whom nearly 2.0 km$^2$ (200 Ha) claimed in 2018.

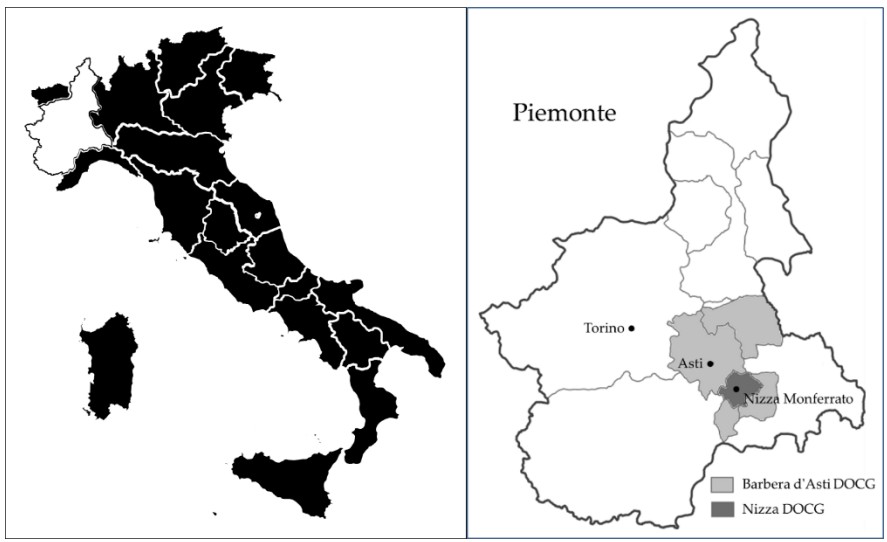

**Figure 1.** Production zones of *Barbera d'Asti/Barbera d'Asti* superiore and *Nizza*.

The main differences between *Barbera d'Asti*, *Barbera d'Asti superiore* and *Nizza* designations are shown in Table 1.

**Table 1.** Differences between *Barbera d'Asti*, *Barbera d'Asti Superiore* and *Nizza* designations.

| Parameter | Barbera d'Asti | Barbera d'Asti Superiore | Nizza |
|---|---|---|---|
| Production zones | 116 communes in the Asti province and 51 communes in the Alessandria province | 116 communes in the Asti province and 51 communes in the Alessandria province | 18 Communes in the Asti province |
| Altitude | not above 650 m a.s.l. | not above 650 m a.s.l. | between 150 and 350 m a.s.l. |
| Exposure | suitable for ensuring suitable ripening of the grapes. North exposure is excluded for new plants | suitable for ensuring suitable ripening of the grapes. North exposure is excluded for new plants | exclusively hilly with exposure from south to south west—south east |
| Alcohol content | 12.00% vol. minimum | 12.50% vol. minimum | 13.00% vol. minimum |
| Ageing | 4 months minimum | 14 months minimum, 6 of whom in wood | 18 months minimum, 6 of whom in wood |
| Minimum total acidity | 4.5 g/L | 4.5 g/L | 5.0 g/L |
| Minimum non-reducing extract | 24.0 g/L | 25.0 g/L | 26.0 g/L |
| Ampelographic composition | Barbera (85% minimum), Freisa, Grignolino and Dolcetto, alone or jointly (15% maximum). | Barbera (85% minimum), Freisa, Grignolino and Dolcetto, alone or jointly (15% maximum). | Barbera 100% |

As it can be seen, specifications in *Nizza* designation are more severe in terms of purity, ageing and zone of production; they were chosen in order to produce wines with recognised higher quality. It is therefore to be expected that *Nizza* is generally considered the finest among the wines obtained from *Barbera* vine; on the Italian market, indeed, *Nizza* is sold at three-fold average prices with respect to *Barbera d'Asti*.

To guarantee producers and consumers, authentication methods must be developed in order to distinguish between *Barbera d'Asti*, *Barbera d'Asti superiore* and *Nizza* wines. Among the different chemical markers available, major and minor elements have been used to distinguish the regionality of

wine [4–6]; another possibility is using trace- and ultra-trace elements as discrimination variables [7–10]. A particular focus must be given on the discrimination power of lanthanides. It is well known their role in providing a link between a specific territory and foodstuffs that originate from it, as a consequence of their homogeneous chemical behaviour, which is not fractionated in the passage between soil, plant and food final product [11–13]. As far as wine is concerned, our previous work [14] and other works suggested that its production chain can cause fractionation of the original soil fingerprint. The role of other trace- and ultra-trace elements is, however, less understood.

Considering that the production zone of *Nizza* is totally contained within that of *Barbera d'Asti* (Figure 1), in this work we wanted to verify whether the distinction between *Nizza*, *Barbera d'Asti superiore* and *Barbera d'Asti,* listed according to their market value from the more expensive to the less one, is possible on the basis of the distribution of trace- and ultra-trace elements. It must be remembered that these wines come from very small areas: 40 km$^2$ (4000 Ha) for *Barbera d'Asti/Barbera d'Asti superiore* DOCG and nearly 2 km$^2$ (200 Ha) for *Nizza* DOCG. ICP elemental analysis and multivariate data analysis were used at the purpose. Samples of wines were from 2015 vintage. Moreover, in order to evaluate the correlation between soil and wine, we analysed samples of soils taken at the various locations of the producers of *Nizza*. The samples of *Barbera d'Asti* and *Barbera d'Asti superiore* were provided by the same producers of *Nizza*, so we can consider that the reference soils are the same. As to the different ampelographic composition of *Barbera d'Asti* and *Barbera d'Asti superiore*, it must be noted that the other grape varieties allowed in addition to Barbera (Freisa, Grignolino, and Dolcetto) are however collected from the same areas.

## 2. Materials and Methods

### 2.1. Materials

High-purity water with resistance > 18 MΩ·cm from a Milli-Q apparatus (Milford, MA, USA) was used in the study. TraceSelect 30% hydrogen peroxide, 69% nitric acid and 37% hydrochloric acid were purchased from Fluka (Milan, Italy). Polypropylene and polystyrene vials, used respectively for sample storage and analysis with an auto-sampler system, were kept in 1% nitric acid and then rinsed with high-purity water when needed. CCS-1 (Rare Earths), CCS-2 (Precious Metals), CCS-4 (Alkali, Alkaline, Non-Transition), CCS-5 (Fluoride Soluble) and CCS-6 (Transition Elements) elements stock solutions (Inorganic Ventures, Christiansburg, VA, USA) at 100 mg/L were used for external calibration; CGIN1 1000 ppm indium solution was used for internal standardisation.

### 2.2. Sample Collection

Soil samples were taken at the producers' locations, collecting one sample for each vineyard. In each place, 1 kg of soil was collected at a depth of 30 cm in order not to include surface contamination.

Wines were obtained directly from each producer (three bottles each); each wine was produced by grapes harvested in single vineyards. The samples were as follows: 9 of *Barbera d'Asti*, 8 of *Barbera d'Asti superiore* and 32 of *Nizza*. Bottles were kept in a cellar and opened only at the moment of analysis.

### 2.3. Sample Treatment

Soil samples were treated according to a standardised procedure [15]: soil was dried at 105 °C for 24 h, after which 1 g was sieved (ϕ 0.2 mm) and extracted with 2 mL of hydrogen peroxide and 8 mL of aqua regia in a microwave oven for 30 min. After centrifugation, the supernatant was withdrawn and the resulting solution was diluted to volume in a 100 mL volumetric flask with high-purity water. Three replicates were measured for every sample solution. The repeatability of the method was checked by analysing five independent aliquots of the same soil sample and resulted to be better than 5% for all elements.

Wine samples were diluted 1:10 with a nitric acid 1% solution containing In 10 ppb as internal standard for the ICP-OES and ICP-MS determination of almost all elements; K, P, S Mg, Ca and Na

were determined on wine samples diluted 1:100 with the same solution. Quality controls were carried out by measuring a calibration solution every 6 samples and verifying that the results were within ±20% error. After opening a bottle, the first 10 mL were discarded in order to avoid contamination from the cork; the leftover wine was then thoroughly mixed before sampling. Care was taken in every manipulation passage, in particular when wine was collected with a micropipette to prepare the diluted solution: this was carried out discarding the first volume collected, so as to avoid contamination from the pipette tip. Three replicates were measured for every sample solution. The repeatability of the method was checked by analysing five independent aliquots of the same wine sample and resulted to be better than 2% for all elements.

## 2.4. ICP-OES Analysis

For major and minor elements, analyses were performed with a Spectro (SPECTRO Analytical Instruments GmbH, Kleve, Germany) Genesis ICP-OES simultaneous spectrometer with axial plasma observation. Instrumental parameters were as follows: RF generator, 40 MHz; RF, 1300 W; plasma power, 1400 W; plasma gas outlet, 12 L/min; auxiliary gas flow rate, 0.80 L/min; nebuliser flow rate, 0.85 L/min; pump speed, 2.0 mL/min. The following elements were determined: Na (589.592 nm), K (766.491 nm), Mg (279.553 nm), Ca (317.933 nm), B (249.773 nm), P (213.618 nm), Si (251.612 nm), Al (396.152 nm) and S (180.731 nm). A multi-element standard solution was prepared starting from Inorganic Ventures (Christiansburg, VA, USA) CCS-4 and CCS-5 multi-element standards containing 100 mg/L for each element; the solution was diluted in order to obtain 10, 5, 1, 0.5, and 0.1 mg/L solutions in 1% nitric acid solution. The limits of detection (LOD) and the limits of quantification (LOQ), calculated respectively as 3 and 10 times the standard deviation of blank measurements [16], are shown in Table 2.

## 2.5. ICP-MS Analysis

For most trace- and ultra-trace elements, analyses were performed with a Thermo Fisher Scientific (Waltham, MA, USA) XSeries 2 model Inductively Coupled Plasma Mass Spectrometer. The instrument is equipped with a quartz torch with silver PlasmaScreen device, a quadrupole mass analyser, a lens ion optics based upon a hexapole design with a chicane ion deflector and a simultaneous detector with real-time multichannel analyser electronics, operating either in analogue signal mode or in pulse counting mode. The inlet system included an ESI PC[3] Peltier Chiller (Elemental Scientific, Omaha, NE, USA) set at +2 °C, incorporating a PFA micro-flow concentric nebuliser and a cyclonic spray chamber. Instrument and accessories are PC controlled by PlasmaLab v. 2.6.2.337 software provided by Thermo Fisher Scientific. Instrument parameters can be found in Aceto et al., 2019 [14].

Measurements were carried out mostly in standard mode. For some analytes the CCT-KED (Cell Collision Technology-Kinetic Energy Discriminator) mode was used to eliminate polyatomic interferences: to do this, an $H_2$/He 8/92% gas mixture was introduced before the quadrupole mass analyser at a flow of 5.00 mL/min. Parameters were as follows: dual mode detection with peak jumping; dwell time 10 ms (standard mode) or 25 ms (CCT-KED mode); 25 sweeps; 3 replicates for a total acquisition time of 60 s.; isotopes used: $^{7}$Li, $^{45}$Sc, $^{49}$Ti, $^{51}$V, $^{52}$Cr, $^{55}$Mn, $^{56}$Fe, $^{59}$Co, $^{60}$Ni, $^{63}$Cu, $^{64}$Zn, $^{75}$As, $^{77}$Se, $^{79}$Br, $^{85}$Rb, $^{88}$Sr, $^{89}$Y, $^{90}$Zr, $^{93}$Nb, $^{97}$Mo, $^{108}$Pd, $^{111}$Cd, $^{120}$Sn, $^{121}$Sb, $^{127}$I, $^{133}$Cs, $^{137}$Ba, $^{139}$La, $^{140}$Ce, $^{141}$Pr, $^{144}$Nd, $^{147}$Sm, $^{153}$Eu, $^{158}$Gd, $^{159}$Tb, $^{163}$Dy, $^{165}$Ho, $^{167}$Er, $^{169}$Tm, $^{174}$Yb, $^{175}$Lu, $^{197}$Au, $^{199}$Hg, $^{205}$Tl, $^{208}$Pb, $^{209}$Bi, $^{232}$Th and $^{238}$U. Among these, V, Cr, Mn, Fe, Co, Ni, Cu, Zn, As, and Bi isotopes were determined in CCT-KED mode. Interferences were evaluated as follows: $CeO^+/Ce^+ < 2\%$ and $Ba^{2+}/Ba^+ < 3\%$. A stability and performance test was performed before each analysis session by monitoring $^{7}$Li, $^{59}$Co, $^{115}$In and $^{238}$U masses and the $^{59}$Co/$^{51}$ClO ratio > 15 for CCT-KED mode. Background signals were monitored at 5 and 220 m/z to perform a sensitivity test on the above-reported analyte masses. The limits of detection (LOD) and the limits of quantification (LOQ), calculated respectively as 3 and 10 times the standard deviation of blank measurements [16], are shown in Table 2.

**Table 2.** LOD and LOQ for the elements determined with ICP-OES and ICP-MS.

| Element | LOD | LOQ | Element | LOD | LOQ | Element | LOD | LOQ |
|---|---|---|---|---|---|---|---|---|
| K [1] | 0.001 mg/L | 0.005 mg/L | Pb [2] | 0.015 µg/L | 0.048 µg/L | Y [2] | 0.3 ng/L | 1.0 ng/L |
| P [1] | 0.062 mg/L | 0.206 mg/L | Ni [2] | 0.060 µg/L | 0.199 µg/L | U [2] | 0.3 ng/L | 1.1 ng/L |
| S [1] | 0.133 mg/L | 0.444 mg/L | Ti [2] | 0.071 µg/L | 0.236 µg/L | Pd [2] | 1.4 ng/L | 4.6 ng/L |
| Mg [1] | 0.004 mg/L | 0.015 mg/L | Cr [2] | 0.061 µg/L | 0.203 µg/L | Cd [2] | 1.4 ng/L | 4.5 ng/L |
| Ca [1] | 0.002 mg/L | 0.007 mg/L | Sc [2] | 6.9 ng/L | 23.0 ng/L | Tl [2] | 0.2 ng/L | 0.5 ng/L |
| Na [1] | 0.007 mg/L | 0.022 mg/L | Li [2] | 5.2 ng/L | 17.2 ng/L | Hg [2] | 8.6 ng/L | 28.5 ng/L |
| Fe [2] | 0.052 µg/L | 0.173 µg/L | Mo [2] | 7.8 ng/L | 26.0 ng/L | Gd [2] | 0.8 ng/L | 2.6 ng/L |
| B [1] | 0.043 mg/L | 0.144 mg/L | Sn [2] | 10.2 ng/L | 34.1 ng/L | Pr [2] | 0.1 ng/L | 0.2 ng/L |
| Si [1] | 0.245 mg/L | 0.816 mg/L | As [2] | 23.5 ng/L | 78.2 ng/L | Sm [2] | 1.2 ng/L | 4.1 ng/L |
| Sr [2] | 0.004 µg/L | 0.014 µg/L | Cs [2] | 0.8 ng/L | 2.8 ng/L | Dy [2] | 0.5 ng/L | 1.6 ng/L |
| Rb [2] | 0.022 µg/L | 0.075 µg/L | Co [2] | 1.3 ng/L | 4.4 ng/L | Th [2] | 0.1 ng/L | 0.2 ng/L |
| Al [1] | 0.006 mg/L | 0.019 mg/L | Zr [2] | 3.3 ng/L | 11.1 ng/L | Yb [2] | 0.3 ng/L | 1.1 ng/L |
| Br [2] | 0.495 µg/L | 1.649 µg/L | Nb [2] | 0.7 ng/L | 2.4 ng/L | Er [2] | 0.4 ng/L | 1.3 ng/L |
| Zn [2] | 0.189 µg/L | 0.630 µg/L | Ce [2] | 3.4 ng/L | 11.5 ng/L | Eu [2] | 0.9 ng/L | 2.9 ng/L |
| Cu [2] | 0.045 µg/L | 0.150 µg/L | Se [2] | 23.7 ng/L | 79.0 ng/L | Bi [2] | 1.4 ng/L | 4.8 ng/L |
| Mn [2] | 0.021 µg/L | 0.070 µg/L | Au [2] | 3.1 ng/L | 10.2 ng/L | Tb [2] | 0.4 ng/L | 1.5 ng/L |
| I [2] | 0.346 µg/L | 1.152 µg/L | Sb [2] | 4.2 ng/L | 13.9 ng/L | Ho [2] | 0.1 ng/L | 0.3 ng/L |
| Ba [2] | 0.072 µg/L | 0.241 µg/L | La [2] | 0.7 ng/L | 2.3 ng/L | Lu [2] | 0.2 ng/L | 0.7 ng/L |
| V [2] | 0.005 µg/L | 0.016 µg/L | Nd [2] | 1.1 ng/L | 3.6 ng/L | Tm [2] | 0.1 ng/L | 0.4 ng/L |

[1] determined by ICP-OES. [2] determined by ICP-MS.

A multi-element standard solution was prepared starting from Inorganic Ventures (Christiansburg, VA, USA) CCS-1, CCS-2, CCS-4, CCS-5 and CCS-6 multi-element standards containing 100 mg/L for each element; this solution was diluted in order to obtain 10, 5, 1, 0.5 and 0.1 µg/L solutions in 1% nitric acid solution. Isotopes responses were corrected by dedicated internal standards using [115]In at 10 µg/L. We used a single element as internal standard because the main aim of this work was to discriminate among wine designations rather than doing measurements with the highest possible accuracy. In addition, a single element standard solution was safer than a multiple elements standard solution from the point of view of possible contamination.

*2.6. Analysis of Certified Samples*

To check performance and recovery of the proposed sample treatment for soil, SRM 2586 (Trace Elements in Soil Containing Lead from Paint) certified material from NIST was analysed according to the described treatment. The results, detailed in Table 3, showed good agreement between the certified and observed concentration values.

It was not possible, however, to have a certified sample for wine.

*2.7. Data Analysis*

Multivariate data analysis was applied to the dataset composed of 57 variables (the elements determined) and 51 samples of wine. Data analysis and graphical representations were performed with XLSTAT v. 2012.2.02 (Addinsoft, Paris, France), a Microsoft Excel add-in software package.

Principal Components Analysis (PCA) was carried out using element concentrations as variables; data were transformed into z-scores before analysis.

For Analysis of Variance (ANOVA), a significance level (or alpha level) of 0.05 was used.

**Table 3.** Certified soil material (Trace Elements in Soil Containing Lead from Paint).

| Element | Certified Values (mg/kg) | Uncertainty | Found (mg/kg) | s.d. |
|---------|--------------------------|-------------|---------------|------|
| Li | 25 [1] | | 74 | 0.60 |
| Sc | 24 [1] | | 11 | 0.04 |
| Ti | 6050 | 660 | 2310 | |
| V | 160 [1] | | 128 | 0.40 |
| Cr | 301 | 45 | 226 | 1.79 |
| Mn | 1000 | 18 | 937 | |
| Fe | 51,610 | 890 | 48,837 | |
| Co | 35 [1] | | 24 | 0.21 |
| Ni | 75 [1] | | 150 | 6.21 |
| Cu | 81 [1] | | 85 | 1.04 |
| Zn | 352 | 16 | 369 | |
| As | 8.7 | 1.5 | [3] | |
| Se | 0.6 [1] | | [3] | |
| Sr | 84.1 | 8.0 | 131.2 | 1.71 |
| Y | 21 [1] | | 19 | 0.16 |
| Nb | 6 [1] | | [3] | |
| Ba | 413 | 18 | 218 | 2.64 |
| La | 29.7 | 4.8 | 27.2 | 0.59 |
| Ce | 58 | 8 | 56.2 | 0.82 |
| Pr | 7.3 [1] | | 7.9 | 0.08 |
| Nd | 26.4 | 2.9 | 29.4 | 0.77 |
| Sm | 6.1 [1] | | 6.0 | 0.11 |
| Eu | 1.5 [1] | | 1.2 | 0.04 |
| Gd | 5.8 [1] | | 6.6 | 0.04 |
| Tb | 0.9 [1] | | 0.9 | 0.02 |
| Dy | 5.4 [1] | | 4.1 | 0.04 |
| Ho | 1.1 [1] | | 0.7 | 0.01 |
| Er | 3.30 [1] | | 2.11 | 0.05 |
| Tm | 0.5 [1] | | 0.3 | 0.01 |
| Yb | 2.64 | 0.51 | 1.68 | 0.03 |
| Lu | [2] | | 0.3 | 0.001 |
| Cd | 2.71 | 0.54 | [3] | |
| Hg | 0.367 | 0.038 | [3] | |
| Pb | 432 | 17 | [3] | |
| Th | 7 [1] | | 14 | 0.10 |

[1] indicative value. [2] not determined in SRM. [3] not determined by us.

## 3. Results and Discussion

Thanks to the relatively low dilution ratio (1:10) and to the use of high purity reagents, it was possible having good results from a large set of analytes. Indeed, concentrations were higher than LOQ for all the analytes indicated in Table 2. All data (ranges in Table 4) resulted to be compatible with the known ranges of elements in wine [17]. The precision was better than 5% for most elements and not lower than 20% even for ultra-trace elements such as heavy lanthanides.

**Table 4.** Ranges of concentration in *Barbera d'Asti* (BA), *Barbera d'Asti superiore* (BAs) and *Nizza* wines.

| mg/L | | BA | BAs | Nizza | µg/L | | BA | BAs | Nizza | ng/L | | BA | BAs | Nizza |
|---|---|---|---|---|---|---|---|---|---|---|---|---|---|---|
| K | ave | 772.7 | 822.7 | 835.2 | Pb | ave | 22.3 | 51.1 | 17.0 | Y | ave | 648 | 466 | 461 |
| | min | 636.7 | 642.9 | 591.0 | | min | 2.67 | 3.64 | 2.26 | | min | 52 | 145 | 72 |
| | max | 908.5 | 1025.9 | 1004.7 | | max | 59.0 | 143.5 | 125.0 | | max | 1995 | 1289 | 1637 |
| P | ave | 210.4 | 238.6 | 249.8 | Ni | ave | 44.3 | 41.8 | 36.6 | U | ave | 502 | 270 | 466 |
| | min | 166.8 | 194.8 | 137.5 | | min | 31.7 | 28.9 | 17.1 | | min | 10 | 56 | 35 |
| | max | 270.0 | 280.7 | 698.8 | | max | 61.9 | 55.7 | 115.9 | | max | 1135 | 415 | 1754 |
| S | ave | 252.9 | 292.1 | 242.8 | Ti | ave | 43.0 | 41.8 | 42.0 | Pd | ave | 86 | 69 | 183 |
| | min | 165.3 | 188.2 | 138.0 | | min | 28.4 | 32.9 | 24.0 | | min | 40 | 50 | 54 |
| | max | 488.1 | 479.9 | 450.1 | | max | 73.1 | 56.4 | 92.4 | | max | 179 | 94 | 1237 |
| Mg | ave | 110.2 | 115.3 | 137.2 | Cr | ave | 16.4 | 24.3 | 18.6 | Cd | ave | 162 | 294 | 191 |
| | min | 88.0 | 98.0 | 93.3 | | min | 9.01 | 8.31 | 10.38 | | min | 94 | 114 | 107 |
| | max | 164.5 | 192.0 | 371.3 | | max | 24.9 | 45.8 | 40.8 | | max | 296 | 901 | 301 |
| Ca | ave | 73.2 | 79.3 | 76.4 | Sc | ave | 40.6 | 42.3 | 40.8 | Tl | ave | 412 | 254 | 306 |
| | min | 54.4 | 60.0 | 55.7 | | min | 39.0 | 40.6 | 30.6 | | min | 252 | 159 | 141 |
| | max | 89.1 | 103.9 | 122.4 | | max | 42.7 | 45.6 | 45.6 | | max | 610 | 352 | 620 |
| Na | ave | 16.25 | 15.69 | 20.31 | Li | ave | 10.3 | 16.0 | 20.7 | Hg | ave | 81 | 86 | 102 |
| | min | 6.79 | 11.05 | 7.84 | | min | 5.40 | 7.75 | 10.77 | | min | 1 | 1 | 1 |
| | max | 41.07 | 20.80 | 44.70 | | max | 14.2 | 26.1 | 37.2 | | max | 376 | 315 | 568 |
| Fe | ave | 1.22 | 3.79 | 0.89 | Mo | ave | 3.58 | 3.15 | 3.67 | Gd | ave | 152 | 97 | 93 |
| | min | 0.34 | 0.58 | 0.04 | | min | 1.15 | 1.87 | 1.41 | | min | 6 | 15 | 7 |
| | max | 1.86 | 14.99 | 4.04 | | max | 10.3 | 5.47 | 16.8 | | max | 541 | 400 | 334 |
| B | ave | 3.51 | 4.05 | 4.51 | Sn | ave | 4.44 | 2.44 | 2.10 | Pr | ave | 143 | 99 | 84 |
| | min | 2.72 | 3.70 | 2.27 | | min | 0.45 | 0.07 | 0.03 | | min | 4 | 12 | 2 |
| | max | 5.17 | 4.53 | 5.91 | | max | 16.5 | 5.80 | 7.55 | | max | 538 | 458 | 317 |
| Si | ave | 3.27 | 3.44 | 3.09 | As | ave | 3.96 | 3.18 | 4.64 | Sm | ave | 131 | 82 | 73 |
| | min | 2.46 | 2.46 | 2.46 | | min | 0.97 | 1.66 | 2.04 | | min | 3 | 16 | 3 |
| | max | 4.61 | 4.46 | 4.90 | | max | 9.57 | 6.63 | 13.9 | | max | 438 | 361 | 283 |
| Sr | ave | 1.10 | 1.37 | 1.53 | Cs | ave | 7.14 | 5.37 | 4.94 | Dy | ave | 112 | 74 | 73 |
| | min | 0.83 | 1.00 | 0.88 | | min | 5.52 | 3.74 | 2.32 | | min | 5 | 19 | 7 |
| | max | 1.35 | 1.70 | 2.43 | | max | 12.9 | 7.44 | 10.5 | | max | 372 | 254 | 263 |

**Table 4.** *Cont.*

| mg/L | | BA | BAs | Nizza | µg/L | | BA | BAs | Nizza | ng/L | | BA | BAs | Nizza |
|---|---|---|---|---|---|---|---|---|---|---|---|---|---|---|
| Rb | ave | 1.42 | 1.16 | 1.16 | Co | ave | 3.59 | 5.20 | 3.64 | Th | ave | 104 | 49 | 80 |
| | min | 1.15 | 0.90 | 0.58 | | min | 2.15 | 3.14 | 1.20 | | min | 4 | 11 | 6 |
| | max | 1.86 | 1.60 | 1.62 | | max | 6.70 | 8.04 | 6.77 | | max | 305 | 133 | 230 |
| Al | ave | 1.13 | 1.09 | 1.19 | Zr | ave | 3.14 | 2.17 | 2.89 | Yb | ave | 73 | 54 | 54 |
| | min | 0.78 | 0.95 | 0.84 | | min | 0.78 | 1.17 | 0.96 | | min | 8 | 18 | 13 |
| | max | 1.69 | 1.22 | 1.79 | | max | 7.90 | 3.41 | 7.25 | | max | 202 | 116 | 191 |
| Br | ave | 0.849 | 0.870 | 0.860 | Nb | ave | 0.74 | 0.14 | 0.47 | Er | ave | 67 | 49 | 48 |
| | min | 0.629 | 0.800 | 0.518 | | min | 0.01 | 0.05 | 0.04 | | min | 5 | 15 | 7 |
| | max | 1.180 | 1.112 | 1.591 | | max | 3.55 | 0.52 | 5.75 | | max | 201 | 129 | 179 |
| Zn | ave | 0.431 | 0.675 | 0.527 | Ce | ave | 1.07 | 0.71 | 0.61 | Eu | ave | 89 | 69 | 77 |
| | min | 0.109 | 0.397 | 0.195 | | min | 0.04 | 0.05 | 0.04 | | min | 48 | 34 | 25 |
| | max | 0.769 | 1.175 | 1.416 | | max | 4.69 | 3.68 | 2.36 | | max | 176 | 131 | 139 |
| Cu | ave | 0.474 | 0.334 | 0.387 | Se | ave | 1.34 | 1.42 | 1.95 | Bi | ave | 10 | 11 | 18 |
| | min | 0.006 | 0.013 | 0.025 | | min | 1.06 | 1.09 | 1.04 | | min | 1 | 1 | 1 |
| | max | 1.132 | 0.648 | 1.067 | | max | 2.68 | 2.08 | 3.54 | | max | 51 | 44 | 92 |
| Mn | ave | 0.249 | 0.236 | 0.328 | Au | ave | 0.12 | 0.18 | 0.34 | Tb | ave | 19 | 12 | 12 |
| | min | 0.036 | 0.036 | 0.036 | | min | 0.06 | 0.09 | 0.01 | | min | 1 | 2 | 1 |
| | max | 0.708 | 0.403 | 0.885 | | max | 0.23 | 0.54 | 2.69 | | max | 71 | 48 | 44 |
| I | ave | 0.330 | 0.343 | 0.358 | Sb | ave | 0.65 | 0.72 | 0.56 | Ho | ave | 21 | 14 | 15 |
| | min | 0.251 | 0.258 | 0.233 | | min | 0.17 | 0.10 | 0.13 | | min | 1 | 4 | 2 |
| | max | 0.418 | 0.463 | 0.506 | | max | 1.44 | 1.89 | 2.46 | | max | 67 | 44 | 55 |
| Ba | ave | 0.154 | 0.126 | 0.149 | La | ave | 0.58 | 0.41 | 0.35 | Lu | ave | 11 | 9 | 9 |
| | min | 0.109 | 0.078 | 0.054 | | min | 0.01 | 0.02 | 0.01 | | min | 1 | 4 | 2 |
| | max | 0.207 | 0.185 | 0.280 | | max | 2.31 | 2.05 | 1.37 | | max | 30 | 19 | 34 |
| V | ave | 0.038 | 0.007 | 0.027 | Nd | ave | 0.59 | 0.43 | 0.34 | Tm | ave | 9 | 7 | 7 |
| | min | 0.000 | 0.001 | 0.000 | | min | 0.02 | 0.07 | 0.01 | | min | 1 | 1 | 1 |
| | max | 0.167 | 0.027 | 0.264 | | max | 2.12 | 1.88 | 1.34 | | max | 29 | 17 | 26 |

In the following sections, we will discuss the possibility of using the elemental distribution, or part of it, to distinguish between *Barbera d'Asti, Barbera d'Asti superiore* and *Nizza* wines. It must be remembered that *Barbera d'Asti* and *Barbera d'Asti superiore* are indeed parts of the same designation, i.e., *Barbera d'Asti* DOCG, therefore they are produced in the same geographic areas; in addition, the territory of *Nizza* designation is totally contained inside that of *Barbera d'Asti*. Therefore, differences among these wines may be expected, rather than from soil, because of oenological practices and, in particular, of ageing (see Table 1).

### 3.1. Lanthanides

Our previous work on the use of lanthanides distribution as traceability markers [14] clearly indicated that the original fingerprinting given by soil is lost during the winemaking process. The same conclusion arose from other past works: Jakubowski et al. [18] in 1999 questioned the fact that rare earth elements (REE) distribution could be considered as reliable fingerprint for the geographic provenance of a wine. Nicolini et al. [19] and Castiñeira et al. [20] both advised that fining treatment with bentonite could lead to fractionation of the original trace element distribution in white wines. Rossano et al. [21] in their study on the influence of clarification, filtration, and storage on the concentration of REE in white wines, found that these processes provided a range of effects ranging from an overall increase to fractionation resulting in small increase of light REEs. As to red wines, Mihucz et al. [22] and Tatár et al. [23] found similar behaviours respectively in Romanian and Hungarian red wines.

The cited studies were mainly focused on the variation of *absolute* concentrations of lanthanides, or on the variation of their distribution along the wine chain without any reference to soil. In the present study we wanted to deepen the relationship between soil and wine, by comparing their distributions after normalisation to Ce according to the formula $[Lanthanide]_{Ce-normalised} = [Lanthanide]_{sample}/[Ce]_{sample}$. Normalisation allows a better comparison between samples (soil and wine) whose concentrations differ by 2–3 orders of magnitude. The lanthanides distributions of all our wine samples follow the Oddo-Harkins rule (Figure 2a, Ce-normalised data for *Nizza* wines, shown in logarithmic scale in order to highlight the differences on the heavy lanthanides that could not be properly appreciated under a linear scale). The behaviour of some lanthanides, however, is apparently unusual. In particular, the content of Nd, Dy, Er and Yb is higher than expected. This cannot be ascribed to isobaric interferences in the determination by ICP-MS: $^{144}$Nd is isobaric with $^{144}$Sm but its interference is automatically subtracted via software and the only known polyatomic interference is from $^{96}Ru^{16}O^+$ [24] which can be safely excluded being the level of Ru in our samples under LOD; $^{163}$Dy has positive interference from $^{147}Sm^{16}O^+$ but $^{147}$Sm accounts for only 15% of total Sm; $^{174}$Yb has positive interference from $^{158}Gd^{16}O^+$ ($^{158}$Gd accounts for 25% of total Gd) but the Gd/Yb ratio is ranging from 0.304 to 3.618, so no correlation seems to exist. The behaviour of $^{167}$Er could be explained in terms of positive interference from $^{151}Eu^{16}O^+$, as $^{151}$Eu has, in turn, interference from $^{135}Ba^{16}O^+$, but no correlation exists indeed between $^{167}$Er and $^{151}Eu^{16}O^+$, nor between $^{167}$Er and $^{135}$Ba.

The behaviour of Eu is widely variable but this is due to the fact that both Eu isotopes, $^{151}$Eu and $^{153}$Eu, suffer from positive interference from Ba oxides ($^{135}Ba^{16}O^+$ and $^{137}Ba^{16}O^+$ respectively); as this interference cannot be resolved with the instrument used in this study (a low resolution quadrupole mass spectrometer), the signal of Eu depends indeed on the content of Ba which is highly variable.

By contrast, the lanthanides distributions determined in the corresponding samples of soil, collected at every location of *Nizza* producers (Figure 2b), are highly homogeneous and closely follow the Oddo-Harkins rule with a general lowering trend of heavy lanthanides. This is the expected behaviour, considering the very small size of the production area of *Nizza*.

To evaluate numerically the different behaviour of lanthanides in wines and soils, as far as Ce-normalised data are concerned, the average RSD (calculated on all lanthanides except Ce) was 55.2% in wines but only 10.0% in soil samples.

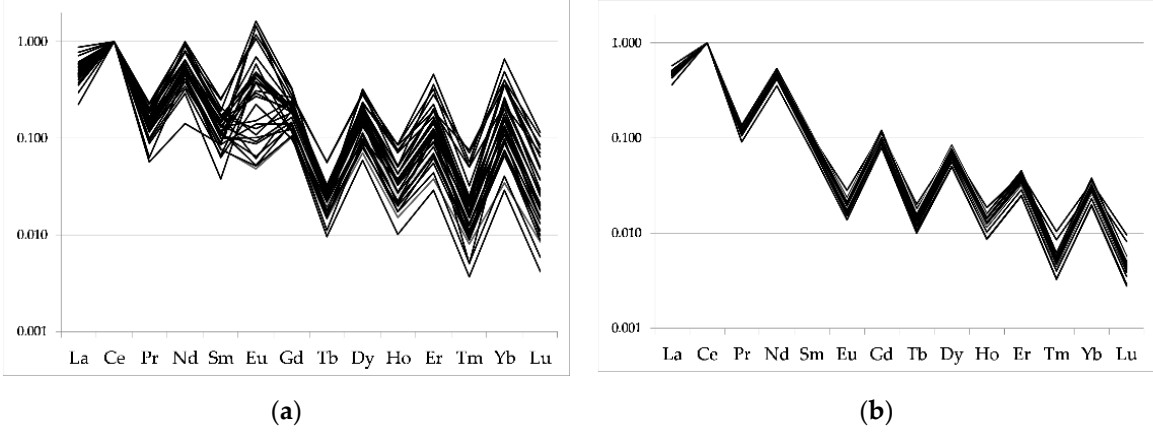

**Figure 2.** Lanthanides distributions in samples of *Nizza* wines (**a**) and in the corresponding soils (**b**). Each line represents 1 bottle (**a**) or 1 soil (**b**).

In the end, it must be accepted the fact that the winemaking processes had heavily influenced the lanthanides distribution, possibly as a consequence of the use of clarifying materials such as bentonite, as it was already cited in our previous work on Moscato d'Asti [25]; bentonites are indeed used by nearly all the producers of *Nizza* wine. According to these results, it is apparent that lanthanides cannot act as traceability markers as they are not representative of the original fingerprint, i.e., the distribution in soil. Not surprisingly, an attempt of distinguishing between *Barbera d'Asti, Barbera d'Asti superiore* and *Nizza* wines on the base of Ce-normalised data of lanthanides, using pattern recognition techniques, was unsuccessful (data not shown).

### 3.2. Comparison between Wines and Soils

It was possible to deepen the knowledge on the behaviour of lanthanides, considering the cases where a winemaker produced two or three designations starting from grapes grown on the same or similar soil. Figure 3 shows some comparisons between wines and corresponding soils (Ce-normalised data, logarithmic scale):

(a)  comparison between one *Barbera d'Asti* and one *Nizza* wine produced from the same vineyard: apparently, they show the same distribution, different from that of the corresponding soil;

(b)  comparison between one *Barbera d'Asti*, one *Barbera d'Asti superiore* and one *Nizza* wine produced from the same vineyard: again, the three wines have the same distribution, different from that of soil;

(c)  comparison between three *Nizza* wines obtained by a producer from grapes cultivated in three different but very close vineyards inside a small area: the three wines are more similar among themselves than to each respective soil;

(d)  comparison between three *Barbera d'Asti superiore* wines and one *Nizza* wine obtained by a producer from grapes cultivated in the same vineyards: the four wines are more similar among themselves than to soil.

The results illustrated above highlight the fact that winemaking, irrespective of vintage and ampelographic composition, is much more important in determining the final lanthanides distribution in wine than the geochemical source.

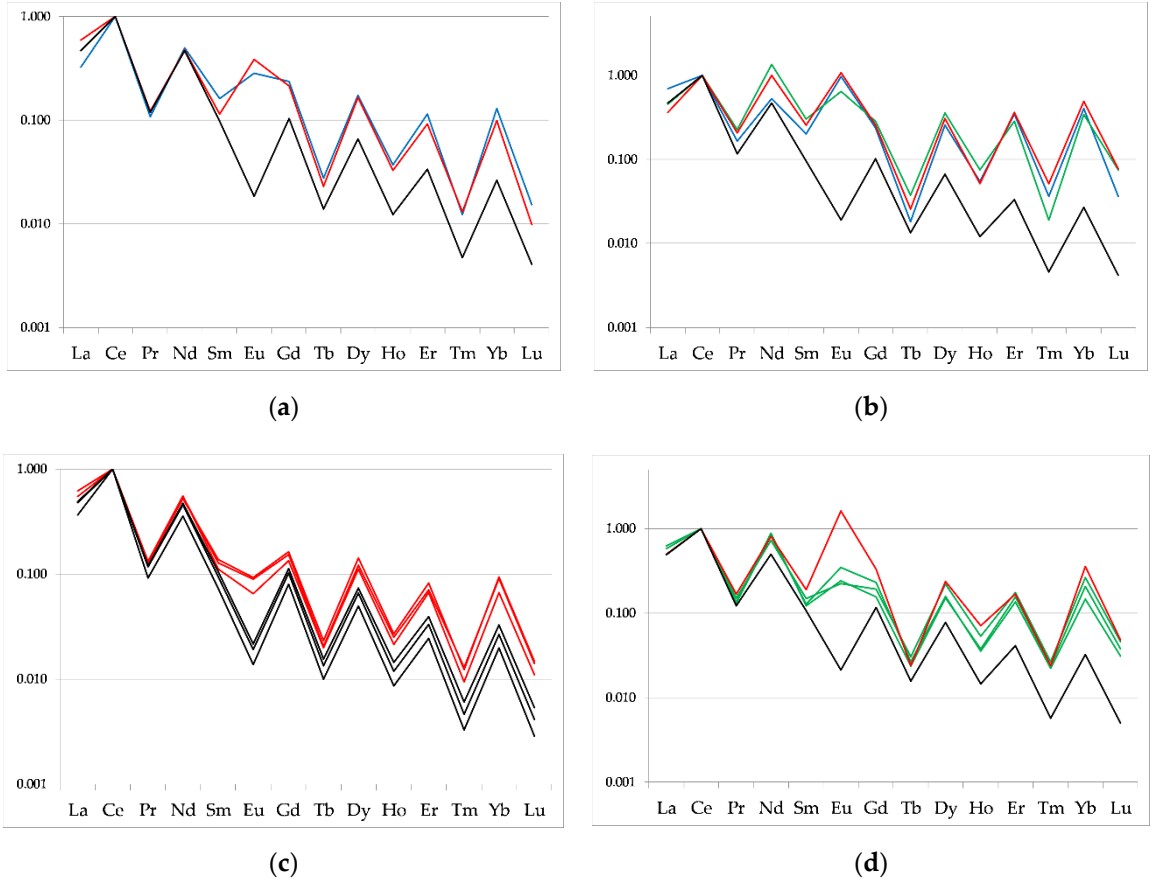

**Figure 3.** Comparison of lanthanides distributions in wines and in the corresponding soils (blue line: *Barbera d'Asti* wine; green line: *Barbera d'Asti superiore* wine; red line: *Nizza* wine; black line: soil) in four cases: (**a**) comparison between one *Barbera d'Asti* and one *Nizza* wine produced from the same vineyard; (**b**) comparison between one *Barbera d'Asti*, one *Barbera d'Asti superiore* and one *Nizza* wine produced from the same vineyard; (**c**) comparison between three *Nizza* wines obtained by a producer from grapes cultivated in three different but very close vineyards inside a small area; (**d**) comparison between three *Barbera d'Asti superiore* wines and one *Nizza* wine obtained by a producer from grapes cultivated in the same vineyards.

## 3.3. Other Trace- and Ultra-Trace Elements

Despite the unsuccessful attempt of using lanthanides to distinguish between *Barbera d'Asti*, *Barbera d'Asti superiore* and *Nizza* wines, we wanted to explore the behaviour of the other trace- and ultra-trace elements. Indeed, many authentication studies on wines generically exploit the whole of trace elements rather than only lanthanides [9,26–28]. Hopfer et al. [29], as an example, were able to classify Californian wines according to their vineyard origin and their processing winery with respect of soil elemental content and viticultural practices.

It is well known that winemaking treatments can affect the mineral content of wine. Clarification with bentonites has strong effects in varying the original metal distribution [30], as already pointed out with reference to lanthanides. Fermentation with different yeast strains markedly affects the content of alkaline, alkaline-earth and transition metals [31]. In a recent study, Catarino et al. [32] followed the trend of elements during winemaking, highlighting the role of the different steps in modifying the original elemental composition in soil.

Pohl reviewed the possible sources of metals [17] in wine, indicating the primary source as the natural contribution from soil, regulated by the climatic condition during grapes growth; a secondary source in the external impurities coming from environment, outside and inside the cellar work; a third source in the oenological practices. Other sources of variation can be the following:

- pH of soil;
- type of rootstock;
- vine growing system;
- type of cultivar;
- time of harvest (it can change from one zone to another and from a farm to another, even at short distances)
- type of collection (manual and/or mechanical)
- Transfer time (from vineyard to cellar) and temperature conditions
- Different types of processing that the product can undergo depending on the objectives of the company grape pressing (time, duration, temperature)
- use of yeasts (usually different from a farm to another)
- duration of maceration and therefore of extraction from skins;
- further processing steps (ageing in steel, barrique—type of wood and provenance—or bottles);
- conservation conditions (temperature, relative humidity, etc.).

Another factor to be considered is of course the thermopluviometric trend, but in this work all wine samples were from the same vintage.

After evaluating the role of lanthanides, in our study we used all the elements determined by ICP-OES and ICP-MS to verify the possibility of discriminating between *Barbera d'Asti, Barbera d'Asti superiore* and *Nizza* wines. The dataset was composed of 57 variables (the elements determined) and 51 samples (wines of the three designations). Principal Components Analysis (PCA) was used; data were transformed into z-scores before analysis. However, no satisfactory results were obtained (data not shown).

Better results were obtained after dividing the samples into two groups, the first containing *Barbera d'Asti* wines and the second containing *Barbera d'Asti superiore* plus *Nizza* wines, i.e., the less aged wines against the more aged ones. A preliminary test by means of Analysis of Variance (ANOVA) indicated that Li, Rb, Sr, B, and Tl were the variables with the higher discriminating power within this scheme (alpha = 0.05). We then carried out PCA analysis using only these five variables: the results of PC1 vs. PC2 score plot (Figure 4), accounting for 70.13% of total variance, suggests that a discrete discrimination is achievable between the younger *Barbera d'Asti* (blue circles in figure) and the more aged *Barbera d'Asti superiore* and *Nizza* wines (red circles in figure).

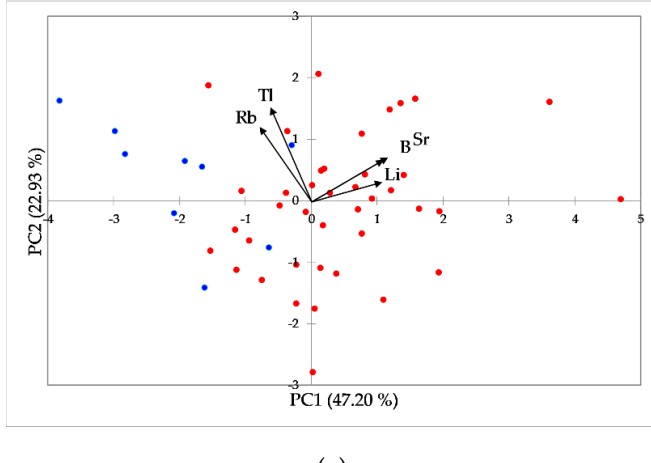
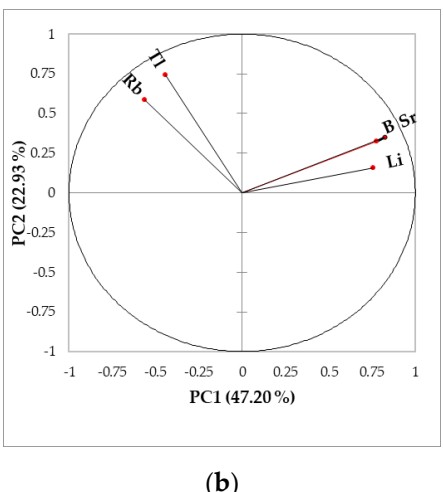

(**a**)  (**b**)

**Figure 4.** (**a**) PC1 vs. PC2 score plot obtained using Li, Rb, Sr, B, and Tl (Blue circles: *Barbera d'Asti* samples; red circles: *Barbera d'Asti superiore* and *Nizza* samples. Black arrows indicate loadings); (**b**) PCA correlation circle.

The information arising from the loadings (black arrows in figure) indicates that *Barbera d'Asti superiore* and *Nizza* wines have a higher content of B, Li and Sr, while *Barbera d'Asti* wines have a higher content of Rb and Tl. Although alkaline and alkaline-earths elements are considered good indicators of geographical origin, in the present study their role must be considered in the light of oenological practises, being the origin of the samples nearly the same or at least too close to be discriminated (it must remembered that the samples of *Barbera d'Asti* and *Barbera d'Asti superiore* analysed in this study come from producers of *Nizza*). Three factors must be considered:

1. The alcoholic content: Catarino et al. [32] showed that the concentration of Rb is inversely proportional to alcohol %, which is in agreement with our data if we consider that the average alcohol % is 14.2 for *Barbera d'Asti* wines and 14.7 for *Barbera d'Asti superiore/Nizza* wines.
2. The widespread use of bentonites by producers of these wines: Catarino et al. [30] showed that this treatment causes a strong fractionation of the original elemental distribution in musts; in particular Li, Sr and Tl were found to increase after bentonites treatment, while B and Rb decreased. However, bentonites are widely used in the production of all *Barbera* designations.
3. The main difference between *Barbera d'Asti* and *Barbera d'Asti superiore/Nizza* is ageing, which involves a more or less prolonged contact with barriques. Kaya et al. [33] studied the effect of wood aging on the mineral composition of wine; Sr was found to increase significantly in wines aged in wood, while for Li, Rb, and Tl no significant effect was registered. These results partially confirm the differences found in our study with concern to Sr, which is higher in *Barbera d'Asti superiore/Nizza* than in *Barbera d'Asti*.

In the end, it is possible that the elemental differences arisen in this study be a combination of all the factors above described. The role of Tl is hard to be explained, considering that this metal must be included in the group of contaminant elements of wine [34]. Even the role of B is still to be accounted for.

## 4. Conclusions

The results obtained from the elemental analysis of *Barbera d'Asti, Barbera d'Asti superiore,* and *Nizza* wines show clearly that the distribution of metals in wine reflect the features of oenological practises rather than the features of soil, in particular with concern to lanthanides. Nevertheless, despite the fact that these three wines are produced in very close if not overlapping areas, it is possible to discriminate the younger *Barbera d'Asti* from the more aged—and more valuable—*Barbera d'Asti superiore* and *Nizza* according to the elemental content, using as chemical descriptors some metals present at trace level concentration, that is Li, Rb, Sr, B, and Tl. These results must be taken as preliminary, however, as only one vintage has been considered, and need confirmation by repeating analysis on at least three vintages.

**Author Contributions:** M.A., F.G. and C.C. conceived and designed the experiments; F.G., E.C. and C.C. performed the experiments; M.A., E.R., M.P. and C.T. analysed the data; M.A. wrote the paper. All authors have read and agreed to the published version of the manuscript.

**Funding:** This research received no external funding.

**Acknowledgments:** The authors are grateful to the producers of *Nizza* who provided samples of wine and in particular the *Associazione dei produttori del Nizza DOCG*.

**Conflicts of Interest:** The authors declare no conflict of interest.

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
