# Peer review of "Authentication and Traceability Study on Barbera d’Asti and Nizza DOCG Wines: The Role of Trace- and Ultra-Trace Elements"

_beverages, doi:10.3390/beverages6040063_

Round 1

Reviewer 1 Report

Generally, this paper investigated the possibility of utilising trace- and ultratrace elements as indicators to distinguish three different Barbera d’Asti designation. The idea proposed by this work could be of interest to local wineries, general consumers and specific quality control authorities, but definitely needs more solid evidence as support.

Here are some specific comments:

  1. The English language of this article needs to be improved a lot before publication, especially the introduction and materials and methods parts. There are too many overloaded and ambiguous long sentences that could be divided into several shorter ones, such as the first (L28 – L44) and third (L51 – L55) paragraphs, and ‘To guarantee producers and consumers, …’ (L56, very confusing). Similar problem is significant for the abstract, and therefore requires rephrase.
  2. Couldn’t see the meaning of including the history of Barbera d’Asti DOCG in the introduction part (L1).
  3. L82: No information about the measured elements is included here. What are they? At what purity? Could be included in a supplementary table.
  4. L85: More detail is required for soil and wine sampling. For each producer, how many bottles of wine sample were collected? All from vintage 2015 or different vintages also involved? From how many different vineyards the grapes were harvested? For each vineyard, how many different soil samples per how many square meters were collected? All of these informations need to be clarified. In particularly, soil samples are generally very heterogeneous. Multiple samples from one vineyard is necessary. One single soil sample doesn’t have enough power to represent the whole vineyard.
  5. L90: Any reference for soil sample preparation? The final solution was liquid only or mixture of liquid and solid soil? If pure liquid, at which step the solid part was excluded?
  6. L97: After discarding the first 10 mL of the wine sample, was the leftover wine mixed before sampling?
  7. L144: the multi-element standard of what elements need to be listed somewhere.
  8. L192: Why element Ce was selected as the standard? Any special property of this element?
  9. L285: The usage of ‘younger’ and ‘aged’ wines is confusing. Are they all from vintage 2015? More proper terminology should be picked.
  10. L285: The ANOVA result, especially for the selected five elements, needs to be included in the article, including the confidence interval.
  11. L288: According to the score plot, only partial discrimination of the two designation was achieved. There is a very significant overlap of samples from the two designations in the middle to little bit left on the figure. Therefore it is not ‘good’ discrimination.
  12. L294: Similar to comment 11, due to the overlap of the samples, approximately 1/3 to 1/2 of the red circle samples (on the negative side along PC1) were supposed to have lower concentrations of B, Sr and Li, and higher concentrations of Tl and Rb, which was opposite to the statement at line 294 – 295.
  13. Figure 4: the correlation between the proposed five elements and the discrimination of the wine samples is not solid enough according to this figure. A PCA correlation circle may show clearer relationship.
  14. L323: Even with the concession statement, as mentioned in comment 11, considering the partial discrimination of the wine samples, more solid evidence is required to give the conclusion of ‘it is possible to discriminate the …’, especially this trial was only conducted with single vintage. If no more evidence could be supplied, the conclusion part needs to be rephrased.

Reviewer 2 Report

Hello authors. Interesting paper. I had some notable questions about your experimental design and statistical analysis. Please find my comments/questions below.

Line edits

17-The end of this sentence was a little confusing. Maybe change to “…which follows the most stringent rules, sells at three times the average price, and is considered to have the highest market value”.

35-Move “also” before “provided” or remove from the sentence.

54-Remove “even.”

78-Add “resistance >18 MΩ·cm” when describing the Milli-Q system

79-Move % before chemical, e.g., “30% hydrogen peroxide.”

203-Remove commas and replace with decimals to match tables in ratio values

Further edits and comments

Introduction

  • Your introduction mostly focused on the ultra-trace elements (rare earth elements) that you worked on, but since you also analyzed major elements, a couple of sentences should be added about other work that has used major and other trace elements to distinguish the regionality of wine.
  • The last sentence of the introduction was confusing. Were all the Barbera d’Asti and Barbera d’Asti superior wines collected from the same plot of land as the Nizza? Or only a subsection? Also, in Table 1, you mention that Barbera d’Asti and Barbera d’Asti superior wines can have different ampelographic composition than the Nizza. Was the fruit all grown in the same vineyard/area? Would you expect this to affect the elemental content, and if so, how can you use the same reference soil samples? Further description could also be moved to the Materials and Methods rather than the Introduction, but these questions should be answered.

Materials and Methods

  • In section 2.2, more description is needed on the wine samples. How many wines from each region/type were collected? Vintage year?
  • In section 2.3, why did you only use 1 ISTD? Were quality controls, such as calibration checks and blanks, used throughout the analysis? If your results for the QCs were okay (within +/-20%), then I think a sentence just explaining the drawbacks of using only one internal standard would be in order.
  • I’m also confused about the need to discard the first 10 mL of the wine. Was there visible residue from the cork in the wine? If the cork had been in contact with the wine for a period of time, wouldn’t the elements diffuse throughout the bottle?
  • Were replicate wine samples analyzed from each bottle? This is essential information needed to see your results' statistical power and should always be given. Replicates/sample size should also be given in the tables.
  • In Table 3, include a percent error column. This would make it easier for the reader to see how close your results were to the SRM.
  • Section 2.7 needs more information. What analyses were run? If ANOVA was performed, what was your significance level? This is typically written as alpha = x.

Results and discussion

  • What do good results mean (line 165)? Was the standard deviation acceptable between analytical replicates?
  • In Table 4, since you are distinguishing between three wine types/regions, I would like to see the data separated for the three regions. It would be best to have the average, min, and max shown for each element in the three wine types. I can see that some elements had large concentration ranges. As a reader, I do not know if these wide ranges were seen in all the wines or inherent to a specific wine.
  • A sentence should be added as to why you Ce-normalized the data.
  • In Figure 2, does each line represent 1 bottle or replicate? Your sample size should be given in the caption. Can you color code the lines for the different wine types?
  • For the plots in Figure 3, I was not sure how you choose which wine to represent? Were the wines used in each plot the only ones that correlated to the soil samples? This will be better understood if the soil samples' questions are clarified (see the last bullet in the introduction section). Also, it seems Eu consistently behaved differently than the other elements in the wine and soil. Can you give reasons as to why that happened?
  • What was your set significance level in the ANOVA, and what is the “higher discriminating power”? You should mention if they were also significant because you talked about and measured lanthanides and other trace elements. I have previously mentioned this in the method and materials section. Here it would be good to add “(alpha = x)” at the end of the sentence when discussing significance.
  • You could try a supervised multivariate technique with the three wine styles. I believe some tests like DA is also available on XLSTAT. Did you try any other techniques? Why did you choose PCA?
  • On line 296, you mention not using the alkali earth metals in further analysis; why? Were they not significant with ANOVA? This is not clear, and it is confusing to readers why you bothered measuring them if you don't mention/use them in any further analysis.
  • For the factors you gave on line 301:
  • We cannot confirm your statement about Rb and alcohol levels because you do not give average values for elements for each wine style, which range in alcohol content. I believe this is another reason for you to change Table 4, as I suggested above.
  • You mentioned Sr, Tl, B, and Rb from the Catarino paper and how their study showed the effects bentonites have on these elements’ concentrations. However, these were also some of the elements you claimed to have the highest discriminating power and used in Figure 4. You said bentonites were used in the production of all the studied wines. Does that mean the bentonites affected the wines here differently? Could this have been the cause of using different bentonites? These questions should be answered. It would be beneficial to cite an additional bentonite study that monitored the changes in elemental profiles due to different brands/types of bentonite usage.

Reviewer 3 Report

This manuscript was written in a vivid fashion, thus readers should be attracted to this publication. It is a high-quality text. Figures might be slightly improved if possible.

Please, carefully check the following: kg (not Kg), km (not Km). Data are mostly fine, but a few of them should be presented with max 3 significant figures (e.g. Table 4: 418, 154, 21.9, etc.).

I am just curious about the table 3. You calculated SD (standard deviation). How did you do that? I mean, which data were used for calculus? Quality check is usually presented in percentages (e.g. plus-minus 10%, deemed as acceptable), i.e. relative SD. Please, explain this.   

Round 2

Reviewer 1 Report

n/a

Author Response

No responses needed

Reviewer 2 Report

Hello authors. Thank you for reading through my edits and answering my questions. There are just two points from my original responses that were not covered in your replies. Please see below.

My original question and your response:

Q1) Materials and Methods – In section 2.3, why did you only use 1 ISTD? Were quality controls, such as calibration checks and blanks, used throughout the analysis? If your results for the QCs were okay (within +/-20%), then I think a sentence just explaining the drawbacks of using only one internal standard would be in order.

A2) A statement has been added in section 2.3 concerning the measurements of QCs.

I would like you to answer the bit about internal standards. Typically, more than one internal standard is used when quantifying as many elements that you did. Please explain in your manuscript why you chose only one ISTD and why In. Add a sentence about the drawbacks of using only one internal standard.

The second original question and your response:

Q1) Results and discussion - What was your set significance level in the ANOVA, and what is the “higher discriminating power”? You should mention if they were also significant because you talked about and measured lanthanides and other trace elements. I have previously mentioned this in the method and materials section. Here it would be good to add “(alpha = x)” at the end of the sentence when discussing significance.

A2) The text has been changed according to the suggestion of the reviewer.

Can you please clarify in the text what "higher discriminating power" is? I'm confused if these were the only significant elements according to the ANOVA, or did they have more discriminating power in terms of the PCA results?
